# SAINT: A Phase I/Expanded Phase II Study Using Safe Amounts of Ipilimumab, Nivolumab and Trabectedin as First-Line Treatment of Advanced Soft Tissue Sarcoma

**DOI:** 10.3390/cancers15030906

**Published:** 2023-01-31

**Authors:** Erlinda Maria Gordon, Sant P. Chawla, Walter Andree Tellez, Elan Younesi, Sonu Thomas, Victoria S. Chua-Alcala, Hripsime Chomoyan, Chrysler Valencia, Don Arlen Brigham, Ania Moradkhani, Doris Quon, Amornchit Srikureja, Steven G. Wong, William Tseng, Noah Federman

**Affiliations:** 1Sarcoma Oncology Research Center, Santa Monica, CA 90403, USA; 2Aveni Foundation, Santa Monica, CA 90403, USA; 3Department of Surgery, Division of Surgical Oncology, City of Hope National Medical Center, Duarte, CA 91010, USA; 4UCLA David Geffen School of Medicine, Los Angeles, CA 90095, USA

**Keywords:** immune checkpoint inhibitor, alkylating agent, soft tissue sarcoma, immunotherapy, chemotherapy, ipilimumab, nivolumab and trabectedin

## Abstract

**Simple Summary:**

The SAINT study is an early-stage clinical trial based on the hypothesis that immunotherapy is more effective when given earlier in the course of the disease in patients with advanced soft tissue sarcoma. It consists of two parts. Phase I aims to determine the maximum acceptable dose of a cancer drug known as trabectedin when combined with two immunotherapy drugs, ipilimumab and nivolumab, in previously treated patients with soft tissue sarcoma. Phase II of the study aims to determine if the combination of ipilimumab, nivolumab and trabectedin improves tumor control and survival of previously untreated patients with advanced soft tissue sarcoma.

**Abstract:**

Background: This Phase 1/2 study is based on the hypothesis that immune checkpoint inhibitors are more effective when given earlier in the course of the disease for advanced soft tissue sarcoma. Methods: Phase I endpoints—maximum tolerated dose in previously treated patients; Phase II endpoints—best response, progression free survival and overall survival and incidence of adverse events in previously untreated patients; Phase I treatments—escalating doses of trabectedin (1.0, 1.2, 1.5 mg/m^2^) as continuous intravenous infusion over 24 h every 3 weeks, 1 mg/kg of ipilimumab given intravenously every 12 weeks, and 3 mg/kg of nivolumab given intravenously every 2 weeks; Phase II treatments—maximum tolerated dose of trabectedin and defined doses of ipilimumab and nivolumab. Results: Phase I (*n* = 9)—the maximum tolerated dose of trabectedin was 1.2 mg/m^2^; Phase II (*n* = 79)—6 complete responses, 14 partial responses, 49 stable disease, 25.3% best response rate, 87.3% disease control rate; median progression-free survival, 6.7 months (CI 95%: 4.4–7.9), median overall survival, 24.6 months (CI 95%: 17.0–.); Grade 3/4 therapy-related adverse events (*n* = 92)—increased ALT (25%), fatigue (8.7%), increased AST (8.7%), decreased neutrophil count (5.4%) and anemia (4.6%). Conclusion: SAINT is a safe and effective first-line treatment for advanced soft tissue sarcoma.

## 1. Introduction

Advanced soft-tissue sarcoma (STS) is associated with a poor prognosis, and treatment options are limited to anthracycline-based chemotherapy, i.e., doxorubicin, administered alone or in combination with alkylating agents (ifosfamide and/or dacarbazine) [1]. Studies from the last 32 years indicate an estimated median survival of 8 to 13 months for first-line treatment [2,3,4] and 2 to 6.6 months for second-line treatment [1].

In 2014, Judson et al. [5] reported on the best results from a phase III randomized multicenter study comparing treatment outcome parameters and the incidence of adverse events in 455 STS patients who received doxorubicin alone or doxorubicin and ifosfamide. There was no significant difference in overall survival (OS) between the groups (median OS 12.8 months in the doxorubicin group vs. 14.3 months in the doxorubicin and ifosfamide group). However, median progression-free survival (PFS) was significantly longer for the doxorubicin and ifosfamide group (7.4 months) than for the doxorubicin group (4.6 months), and overall response (OR) was greater in the doxorubicin and ifosfamide group vs. the doxorubicin group (26% vs. 14%, respectively). More Grade 3 or greater treatment-related adverse events (TRAEs) were reported in the doxorubicin plus ifosfamide group compared with the doxorubicin alone group.

In 2018, Nagar [6] described real-world treatment patterns and outcomes for patients with advanced STS not amenable to surgery or radiotherapy in the United Kingdom, Spain, Germany, and France. In 807 patients, the most common first-line regimens were doxorubicin alone (41%), doxorubicin plus ifosfamide (19%), docetaxel plus gemcitabine (9%), paclitaxel alone (4%), and ifosfamide (4%). The median OS was 17.6 months. The authors concluded that new therapies that improve OS in advanced STS are needed. Recently, several agents received approval for the treatment of advanced STS, including pazopanib, trabectedin and eribulin [7,8,9]. Although these agents have marginally improved PFS and OS, they rarely resulted in a full cure [10,11].

Global collaboration remains key to confirming concepts in controlled, randomized clinical trials, and molecular profiling may likely form the basis of standard treatments in the near future [12]. However, inventions arise from thinking “outside the box” and these ideas could be tested in single center studies before further evaluation in multicenter, controlled, randomized clinical studies.

In recent years, the use of immune checkpoint inhibitors (ICIs) in advanced sarcoma has gained increasing popularity in the medical and scientific community [13,14]. Immune checkpoint inhibitors are approved by the United States Food and Drug Administration (USFDA) for the treatment of various clinical indications other than STS. They include ipilimumab, a human cytotoxic T-lymphocyte antigen 4 (CTLA-4)-blocking antibody for unresectable or metastatic melanoma and as adjuvant therapy for cutaneous melanoma [15]. Nivolumab is a programmed death receptor-1 (PD-1) blocking antibody approved for several indications in solid tumors and hematologic malignancies [16]. Trabectedin is a marine-derived alkaloid indicated for treatment of anthracycline-resistant advanced liposarcoma (LPS) or leiomyosarcoma (LMS) [17]. Trabectedin not only destroys cancer cells and exposes the tumor neoantigens for immune recognition, but also destroys growth promoting M2 macrophages in the tumor microenvironment [13].

The aim of SAINT (A phase I/expanded phase II study using safe amounts of ipilimumab, nivolumab and trabectedin as first-line therapy for soft tissue sarcoma (NCT03138161)) is to determine if this combination regimen is a safe and effective first-line treatment for advanced STS. The guiding hypothesis is that sarcoma cells are most immunogenic earlier in the course of the disease before cancer immunoediting occurs [18]. Figure 1 is a graphic illustration of the mechanisms of action of ipilimumab, nivolumab and trabectedin in the tumor microenvironment.

## 2. Materials and Methods

The primary objective is to evaluate the safety, dose-limiting toxicity (DLT), and maximum tolerated dose (MTD) of trabectedin in combination with defined doses of ipilimumab and nivolumab using the cohort of three design [19]. The secondary objectives are to evaluate the best objective response by Response Evaluation Criteria in Solid Tumors (RECIST) v1.1 [20] via computerized tomography (CT) scan or magnetic resonance imaging (MRI) at week 6 and every 6 weeks thereafter until end of treatment and to determine progression-free survival (PFS) at 6 months and overall survival (OS). The exploratory objectives are to correlate RECIST v1.1 with iRECIST [21], treatment outcome parameters with programmed death-1 (PD-1)/programmed death-ligand 1 (PD-L1) expression and other oncogenic drivers in patients’ archived tumors and to correlate absolute lymphocyte count with dexamethasone administration, which is given to prevent hepatotoxicity associated with trabectedin.

Patients of any gender who were 18 years old or older with a diagnosis of locally advanced unresectable or metastatic STS, confirmed histopathologically, and with at least one target lesion ≥1 cm were enrolled. Phase I of the study included previously treated patients, and Phase II enrolled previously untreated patients. The relevant inclusion criteria included adequate hematologic and organ function, and a European Cooperative Oncology Group score (ECOG) of 0–1. The relevant exclusion criteria included previous treatment with an immune checkpoint inhibitor or history of an immune disorder.

Study Design: Phase I of the study involved escalating doses of trabectedin using Storer’s cohort of three design [19] with the MTD as the highest dose level wherein only one patient developed DLT and with two patients having DLT at the next higher dose level. Dose escalation within groups was prohibited. Treatment consisted of escalating doses of trabectedin (1.0, 1.2, 1.5 mg/m^2^) as continuous intravenous infusion over 24 h every 3 weeks, ipilimumab at 1 mg/kg intravenously every 12 weeks and nivolumab at 3 mg/kg intravenously every 2 weeks.

For the Phase II of the study, an additional 70–90 untreated patients were given trabectedin at the maximum tolerated dose and fixed doses of ipilimumab and nivolumab to evaluate antitumor activity and safety of the SAINT regimen in a larger population of advanced STS. The treatment was discontinued when clinical disease progression or significant toxicity occurred. The clinical protocol allowed the principal investigator to determine if surgical resection or biopsy of the tumor was appropriate after several treatment cycles.

Study Duration: The end of the study was defined as either the date of the last visit of the last patient to complete the study or the date of receipt of the last data point from the last patient that was required for primary, secondary, and/or exploratory analysis, as pre-specified in the protocol.

Participant Duration: The treatment ended on the date when the patient received the last dose of ipilimumab, nivolumab and trabectedin. All the patients who at the end-of-study visit had at least one Grade 2 or higher AE or serious adverse events (SAE) were followed for 30 days longer. The patients who completed the study period of one year were placed in a follow-up group and contacted every 6 months to capture unexpected safety events, history of disease progression and to ascertain survival for up to 5 years. All consenting patients were followed for the duration of their survival after the treatment ended.

Safety (DLT, MTD, incidence and severity of adverse events and significant laboratory abnormalities) was the primary endpoint. A safety analysis was performed on all patients who were treated with at least one dose of the study drugs. The incidence of all TRAEs was based on NCI CTCAE v4.03 and was reported in tables according to the severity grade and drug relatedness [22].

For ipilimumab, nivolumab and trabectedin exposure, the total number of doses for each drug received was reported.

The efficacy endpoints, best overall response (BOR), disease control rate (DCR), and PFS were determined by a local radiologic assessment using RECIST v1.1. After the end of treatment, the patients were followed for overall survival every 12 weeks (±3 weeks), or more frequently as needed, until death, withdrawal of consent, or the study closed, whichever came first.

### Statistical Considerations

A clinical study report was generated and updated once the last data point was collected from the last patient that was required for primary, secondary, and/or exploratory analysis, as pre-specified in the protocol.

Baseline Descriptive Statistics: Demographics, age, ethnicity, subtypes of sarcoma, number of patients, patients with locally advanced or metastatic, resectable or unresectable tumors, and ECOG scores are described using descriptive statistics.

Analysis of primary endpoint: The primary endpoint (MTD) was analyzed on the Intention-to-Treat (ITT) in the Phase I of the study. The secondary endpoints were analyzed on the Modified Intention-to-Treat (mITT) populations participating in the expanded Phase II of the study. The ITT population consisted of all subjects who received at least one dose of ipilimumab, nivolumab and trabectedin. The mITT population included all patients who had completed the first cycle of ipilimumab, nivolumab and trabectedin, and had a CT scan or MRI.

Analysis of Secondary Endpoints (Phase II of study): The RECIST v1.1 criteria [20] were used to assess the best overall response (complete response (CR), partial response (PR)), disease control (CR, PR, stable disease (SD)) or progressive disease (PD). The STATA software was used to calculate OS and PFS. The time to event endpoints were summarized descriptively using the Kaplan–Meier (KM) method [23]. The number of patients censored on the cut-off-date and the number of events were provided. Kaplan–Meier estimates were also presented graphically. Data visualization aids were prepared using Microsoft Excel.

Analysis of Exploratory Endpoints The analyses of the exploratory endpoints were descriptive and hypothesis generating in nature.

Ethical Statement: This clinical trial was conducted in accordance with Good Clinical Practice (GCP) as required by the International Council for Harmonisation (ICH) guidelines and in accordance with local laws. The trial was granted approval by the Western Institutional Review Board. All the patients signed the informed consent prior to any study related procedure being conducted and their inclusion in this study. This trial was registered with clinicaltrials.gov, number NCT03138161.

## 3. Results

Enrollment and demographics: Ninety-two patients were enrolled from 19 June 2017 to 3 February 2021. The number of patients studied was 9 patients in Phase I and 92 patients in Phase II. Table 1 shows the baseline characteristics of the patients enrolled according to age group and sex. There were 44 men and 57 women enrolled. There were 8 patients (7.9%) aged between 18 and 28 years, 12 (11.9%) aged 29–39 years, 13 (12.9%) aged 40–50 years, 30 (29.7%) aged 51–61 years, 26 (25.7%) aged 62–72 years, and 12 (11.9%) between the ages of 73 and 83 years. The histologic subtypes of the subjects enrolled in the study are also shown in Table 1: 26 (25.5%) had leiomyosarcoma (LMS), 14 (13.7%) liposarcoma (LPS), 9 (8.8%) undifferentiated pleomorphic sarcoma (UPS), 7 (6.9%) rhabdomyosarcoma, 5 (4.9%) synovial sarcoma and 24 (23.8%) had less than 4% of other histologic subtypes.

Summary of safety analysis: In Phase I, DLT was reported in one of six patients treated at Dose Level II (1.2 mg/m^2^ trabectedin) and consisted of a decreased platelet count with bleeding. The MTD of trabectedin was then determined at Dose Level II. Table 2 shows the TRAEs in Phase I and II according to the drug attribution and severity grade. A total of 49 out of 101 ITT patients (48.5%) had Grade ≥ 3 TRAEs. The Grade ≥ 3 TRAEs related to trabectedin included both hematologic and non-hematologic toxicities, such as fatigue, nausea, fever, exhaustion, dehydration, asthenia, port-site cellulitis, anemia, neutropenia, thrombocytopenia, elevated ALT, elevated AST, and elevated CK. There was a greater incidence of Grade ≥ 3 increased AST, ALT, and CK related to trabectedin than those related to nivolumab or ipilimumab. There were no hematologic toxicities related to nivolumab or ipilimumab. Grade 3–4 TRAEs related to nivolumab include decreased TSH, increased T4, increased TSH, increased alkaline phosphatase, increased AST, increased ALT, hyponatremia, dehydration, pruritus, and psoriasis. Grade 3–4 TRAEs related to ipilimumab included decreased TSH, increased TSH, vomiting, exhaustion, increased AST, increased ALT, increased alkaline phosphatase, hyponatremia, dehydration, and psoriasis. There was a greater incidence of skin and metabolic disorders related to nivolumab or ipilimumab than those related to trabectedin. There were no unexpected adverse reactions, no alopecia nor cardiac toxicity reported in 101 patients. Appendix A shows the ≥Grade 3 adverse events unrelated to the study drugs. At the data cut-off date, 45 (44.6%) patients had died of disease progression.

Summary of efficacy analysis: As per the protocol, an efficacy analysis was performed on the mITT cohort of the expanded Phase II group, i.e., on patients who completed at least one treatment cycle and had a CT scan on Week 6. Of the 92 patients enrolled in the expanded Phase II, 79 patients were included in this mITT cohort. Fourteen patients in Phase II of the study could not be evaluated. Eight patients (8.7%) did not return for treatment after the first dose of trabectedin and were lost to follow up, 3 (3.3%) were admitted to the hospital before completing the first cycle, 2 (2.2%) voluntarily withdrew from study, and 1 (1.1%) discontinued treatment due to an adverse event. Treatment was discontinued for four patients (4.3%) due to worsening psoriasis not revealed during the screening period (*n* = 1), neutropenic fever (*n* = 1), fatigue and intractable vomiting (*n* = 1) and adrenal insufficiency. The median time to follow up for patients included in the efficacy analysis was 18.1 months. The risk per person-time analysis was 25.3 deaths per 1000 person-months. Moreover, we calculated a risk of 93.2 cases of PD per 1000 person-months.

As shown in Table 3, the best overall responses were 6 CR, 14 PR, 49 SD and 10 PD with 25.3% best overall response rate (BORR) and 87.3% DCR. The median PFS was 6.7 months (94% CI: 4.4–7.9) and the median OS was 24.6 months (95% CI: 17.0–.). At 6 months of follow up, the PFS rate was 53.2% and the OS rate was 89.9%. For the ITT population (*n* = 92), the median OS was 19.3 months (95% CI: 13.9–31.5), and the 6-month OS rate was 79.2%. Figure 2 shows the KM curves for OS and PFS from the time of treatment initiation for patients in the expanded Phase II part of the study with numbers at risk and numbers censored in parentheses. Appendix A lists all the patients who participated in Phase I and expanded Phase II by number, histologic subtype, genetic mutations, response, number of specified drug infusions, and PFS and OS for the ITT population.

### Correlative Analysis

*Correlation between RECISTv1.1 and iRECIST:* We correlated the status according to RECISTv1.1 and iRECIST using Pearson correlation taking the coding for each classification. The Pearson correlation coefficient was 1.00 (*p* < 0.0001), which represents a perfect correlation. Moreover, the ORR and DCR were identical when correlating them to RECISTv1.1 vs. iRECIST.

*Correlation between PD-L1 expression and treatment outcome parameters:* PD-L1 positivity was not associated with improved PFS (Hazard Ratio = 0.83; 95% CI: 0.38 to 1.83) nor prolonged OS (Hazard Ratio = 0.48; 95% CI: 0.14 to 1.56).

*Correlation between tumor suppressor TP53 loss/mutation, oncogenic driver MDM2 and treatment outcome parameters:* The positivity to TP53 loss/mutation was not associated with PFS (Hazard Ratio = 0.85, 95% CI: 0.44 to 1.61) nor OS (Hazard Ratio = 0.62, 95% CI: 0.24 to 1.59). Similarly, the MDM2 oncogene was not associated with any significant treatment outcome parameter (OS: HR = 0.75, 95% CI: 0.26 to 2.10; PFS: HR = 0.66, 95% CI: 0.30 to 1.45).

*Correlation between Absolute Lymphocyte Count (ALC) and Dexamethasone treatment:* ANOVA was used to analyze the mean difference in ALC percent change from week to week per patient during the treatment period in 11 patients. There was no significant difference in the mean ALC percent change from week to week per patient during the treatment period (*p* > 0.05).

## 4. Discussion

Immune check point inhibitors have become a mainstay of therapy for melanoma and are currently being developed for various solid tumors [24]. The underlying principle is to thwart the defenses (checkpoints) that tumors utilize to evade the immune system. Immune checkpoint inhibitors have shown some activity in certain types of sarcomas [25,26]. However, the use of immune checkpoint inhibitors has not definitively been shown to work in soft tissue sarcomas, possibly because treatment is given after standard chemotherapy/radiation therapy when the cancer cells have undergone immunoediting and may no longer be immunogenic. Understanding the bifunctional role that the immune system plays in tumor eradication vs. growth promotion is critical in the design and timing of tumoricidal and immunologic therapies for sarcomas [27]. Sarcoma cells are most immunogenic earlier in the course of the disease when the immune system can recognize and destroy them. However, during a period of dormancy, regulatory T cells (T regs) play an immunosuppressive role, allowing cancer cells to mutate and edit themselves (a process known as cancer immunoediting) to become less immunogenic and, hence, escape immune surveillance, resulting in a resurgence of cancer cells that are resistant to therapy [18].

We hypothesized that immune checkpoint inhibitors such as ipilimumab and nivolumab, that promote sustained T cell activation by suppressing T regs [24] would be most effective when given as first-line therapy together with a tumoricidal agent, such as trabectedin, whose plausible mechanism of action is not only to destroy the cancer cells and expose the tumor neoantigens for immune recognition, but also to destroy growth promoting factors in the tumor microenvironment, favoring polarization of M2 (tumor growth-promoting) to M1 (tumor growth-suppressing) macrophages that can directly lyse tumor cells, secrete TNF and IL-12, present antigens, and activate T cells [13]. Moreover, the use of trabectedin is not associated with cumulative toxicity, allowing prolonged administration [28].

Multiple assessments of potential biomarkers such as PD-1/PD-L1, LAG-3 or TIM-3 expression in sarcomas have been inconclusive when correlating them with treatment outcome [29]. Our study shows PD-1/PD-L1 expression in tumors was not associated with improved treatment outcome parameters.

The SAINT study demonstrates the positive impact of immunotherapy on BOR, PFS and OS in patients with advanced STS when given early in the course of the disease. This is independent of PD-1/PD-L1 expression and mutation/amplification of oncogenic drivers in tumors. By indirect comparison, the ORR, PFS and OS appear to be better than those achieved with standard first-line therapy [5,30], with a median PFS of 4.6–5.5 months using doxorubicin alone, and OS appears to be better than that achieved with first-line doxorubicin with or without ifosfamide [5]. Further, by indirect comparison, the SAINT protocol appears to be safer than doxorubicin with or without ifosfamide [5,30]. This is a phase I/II clinical trial with a non-randomized expanded phase II. Therefore, more evidence would be needed from phase III randomized studies to reach definitive conclusions about the superiority or non-superiority of the SAINT regimen over standard chemotherapy as first-line treatment for advanced unresectable or metastatic STS.

## 5. Conclusions

Taken together, the data indicate the following: (1) The primary endpoint was reached. Trabectedin (given at the recommended Phase II dose of 1.2 mg/m^2^) may be safely combined with the immune checkpoint inhibitors ipilimumab (given at the standard dose of 1 mg/kg/dose every 12 weeks) and nivolumab (given at a standard dose of 3 mg/kg every 2 weeks) without dose-limiting toxicity; (2) The incidence of Grade 3 or higher adverse events was as expected, and there were no unexpected adverse events associated with triple therapy using ipilimumab, nivolumab and trabectedin at the defined doses; (3) The best responders (CR and PR) were patients with leiomyosarcoma, liposarcoma, pleomorphic sarcoma, synovial sarcoma, myxofibrosarcoma, endometrial stromal sarcoma and clear cell sarcoma; (4) The RECIST v1.1 was highly correlated with iRECIST; (5) There was no correlation between treatment outcome parameters and PD-L1 expression or mutation/amplification of oncogenic drivers found in patients’ archived tumors; (6) Dexamethasone administration given with Trabectedin every 3 weeks was not associated with a significant reduction in ALCs; (7) The SAINT regimen using safe amounts of ipilimumab, nivolumab and trabectedin is safe and effective as a first-line therapy for advanced soft tissue sarcoma; and (8) Randomized phase III studies are needed to determine the superiority or non-superiority of the SAINT regimen vs. doxorubicin plus ifosfamide as first-line treatment of advanced soft tissue sarcoma.

## Figures and Tables

**Figure 1 cancers-15-00906-f001:**
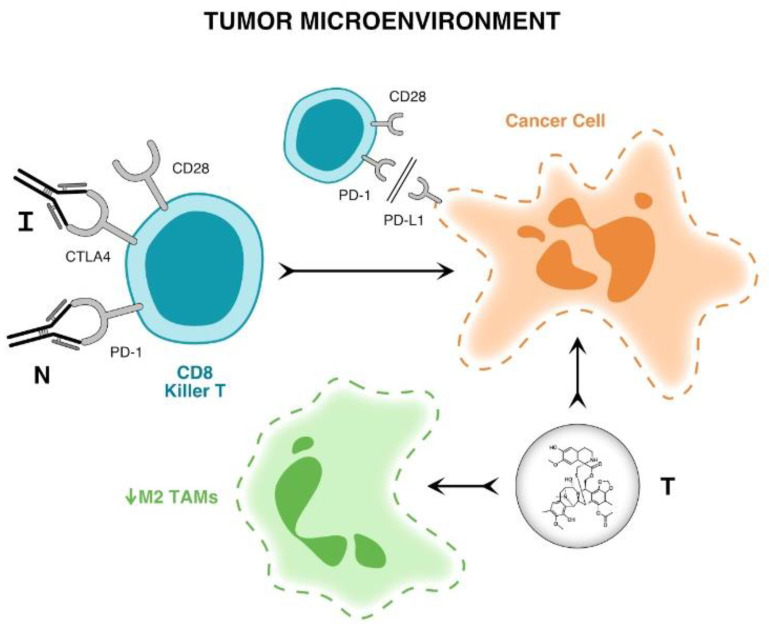
Graphic illustration of the mechanism of action of Ipilimumab, Nivolumab and Trabectedin in the tumor microenvironment. Ipilimumab (I) blocks the CTLA 4 receptor and nivolumab (N) blocks the PD-1 receptor on T cells which then block the PD1- PD-L1 interaction on tumor cells. Trabectedin (T) not only kills cancer cells but depletes the tumor microenvironment of growth promoting or M2 macrophages. The combined effect of these three drugs is sustained T cell activation, resulting in tumor lysis, inhibition of tumor growth or tumor eradication.

**Figure 2 cancers-15-00906-f002:**
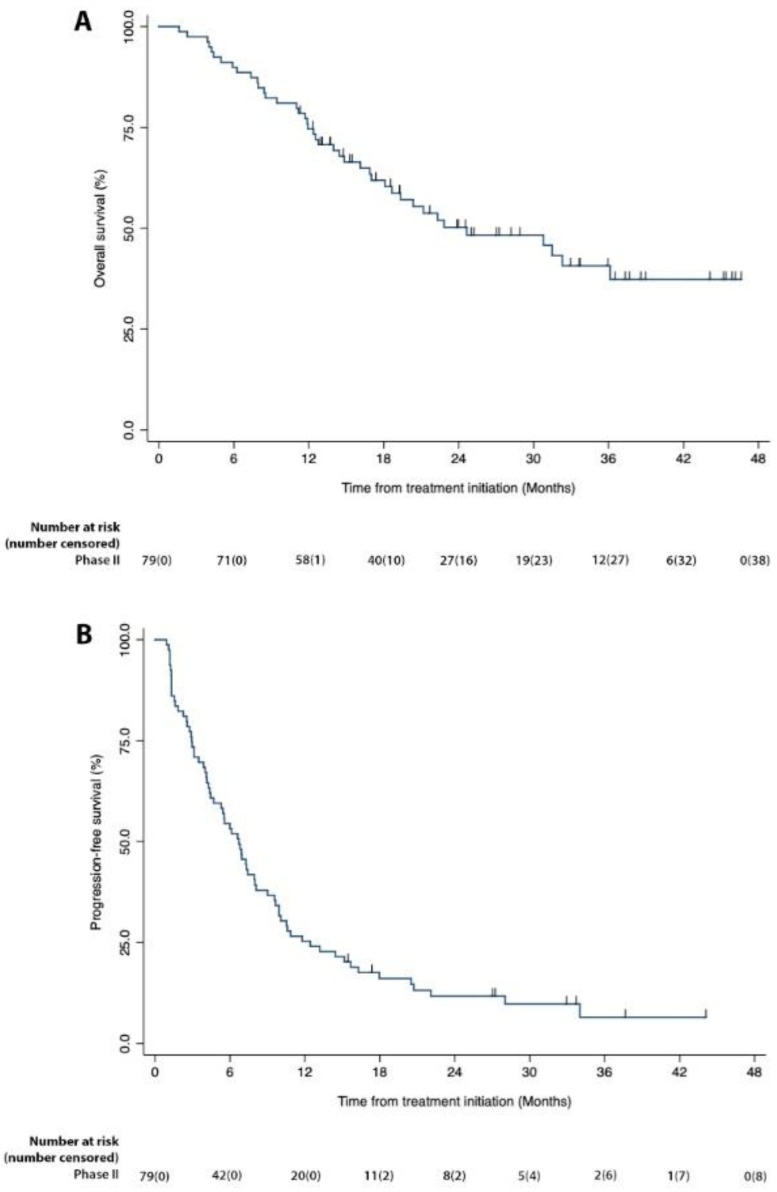
Kaplan–Meier graphs of overall survival and progression free survival for the mITT patients in Phase II of the study. (**A**,**B**) Percent survival is plotted on the vertical axis as a function of time from treatment initiation (months) plotted on the horizontal axis.

**Table 1 cancers-15-00906-t001:** Baseline characteristics.

Patients	*n* = 101
Age	
18–28	8 (7.9%)
29–39	12 (11.9%)
40–50	13 (12.9%)
51–61	30 (29.7%)
62–72	26 (25.7%
73–83	12 (11.9%)
Sex	
Men	44 (43.6%)
Women	57 (56.4%
ECOG Score	
≤1	101 (100%)
Histological type	
Liposarcoma	14 (13.7%)
Leiomyosarcoma	26 (25.5%)
Undifferentiated pleomorphic sarcoma	9 (8.8%)
Rhabdomyosarcoma	7 (6.9%)
Synovial sarcoma	5 (4.9%)
Clear cell sarcoma	4 (3.9%)
Pleomorphic sarcoma	4 (3.9%)
Myxofibrosarcoma	4 (3.9%)
Peripheral nerve sheath tumor	3 (2.9%)
Myxoid liposarcoma	3 (2.9%)
Carcinosarcoma	2 (2.0%)
Desmoplastic small round cell tumor	2 (2.0%)
NOS sarcoma	2 (2.0%)

**Table 2 cancers-15-00906-t002:** ≥Grade 3 Adverse Events Related to Study Therapy.

Phase I—Dose Level 1 (*n* = 3)						
Adverse Event	Trabectedin		Nivolumab		Ipilimumab	
	3	4	3	4	3	4
Investigations						
TSH Increased			1 (33.3%)		1 (33.3%)	
Phase I—Dose Level 2 (*n* = 6)						
Adverse Event	Trabectedin		Nivolumab		Ipilimumab	
	3	4	3	4	3	4
Blood and lymphatic system disorders						
Anemia	2 (33.3%)					
General disorders and administration site conditions						
Fatigue	1 (16.7%)		1 (16.7%)			
Investigations						
Alanine aminotransferase increased	2 (33.3%)		2 (33.3%)			
Platelet count decreased		1 (16.7%)				
TSH decreased			1 (16.7%)		1 (16.7%)	
T4 increased			1 (16.7%)			
Aspartate aminotransferase increased	1 (16.7%)		1 (16.7%)			
TSH increased			3 (50%)		2 (33.3%)	
CPK increased	2 (33.3%)					
Alkaline phosphatase increased	1 (16.7%)		1 (16.7%)			
Musculoskeletal and connective tissue disorders						
Asthenia	1 (16.7%)					
Expanded Phase II (*n* = 92)						
Adverse Event	Trabectedin		Nivolumab		Ipilimumab	
	3	4	3	4	3	4
Blood and lymphatic system disorders						
Anemia	7 (7.6%)	1 (1.1%)				
Gastrointestinal disorders						
Nausea	1 (1.1%)					
Vomiting					1 (1.1%)	
General disorders and administration site conditions						
Fatigue	8 (8.7%)					
Fever	2 (2.2%)					
Exhaustion	1 (1.1%)				1 (1.1%)	
Infections and infestations						
Cellulitis, port-a-catheter	2 (2.2%)					
Investigations						
Aspartate aminotransferase increased	8 (8.7%)	2 (2.2%)	2 (2.2%)		1 (1.1%)	
Alanine aminotransferase increased	23 (25%)	3 (3.3%)	5 (5.4%)		3 (3.3%)	
TSH increased			1 (1.1%)		1 (1.1%)	
Neutrophil count decreased	5 (5.4%)	1 (1.1%)				
Platelet count decreased	2 (2.2%)	2 (2.2%)				
Alkaline phosphatase increased			1 (1.1%)		1 (1.1%)	
CPK increased	2 (2.2%)	2 (2.2%)		1 (1.1%)		
White blood cell decreased	1 (1.1%)					
Metabolism and nutrition disorders						
Hyponatremia			4 (4.3%)		2 (2.2%)	
Dehydration	1 (1.1%)		1 (1.1%)		1 (1.1%)	
Musculoskeletal and connective tissue disorders						
Asthenia	1 (1.1%)					
Skin and subcutaneous tissue disorders						
Pruritus			1 (1.1%)			
Psoriasis			1 (1.1%)		1 (1.1%)	

**Table 3 cancers-15-00906-t003:** Expanded Phase II: Responses to treatment for the mITT population (*n* = 79).

Best Response	Disease Control Rate	Median OS Months (Range) [CI]	Median PFS Months (Range) [CI]	6-Month OS Rate	6-Month PFS Rate
6 CR, 14 PR, 49 SD, 10 PD (25.3% ORR)	87.3%	24.6 (1.6–46.5) (CI 95%: 17.0–.)	6.7 (0.9–44.0) (CI 95%: 4.4–7.9)	89.9%	53.2%

Note: Two patients with PR and one patient with SD had surgical CRs.

## Data Availability

The data presented in this study are available on request from the corresponding author.

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
