# Peer review of "SAINT: A Phase I/Expanded Phase II Study Using Safe Amounts of Ipilimumab, Nivolumab and Trabectedin as First-Line Treatment of Advanced Soft Tissue Sarcoma"

_cancers, 2023, doi:10.3390/cancers15030906_

Round 1
Reviewer 1 Report
congrats for the paper.
page 1, line 36: I think you mean untreated pts in phase II study.
the combination you propose certainly deserves to be published.
are you studying T-regs subpopulations, like Th1? a lab in Milan is having interesting observation on that, on very few numbers in STS at the moment. I'll suggest them to read your paper.
Author Response
Response to Reviewer 1’s Comments
- Page 1, line 36: I think you mean untreated pts in phase II study.
Response: Yes, I meant previously “untreated” patients (See Line 36).
- The combination you propose certainly deserves to be published.
Response: Thank you for your recommendation.
- Are you studying T-regs subpopulations, like Th1? a lab in Milan is having interesting observation on that, on very few numbers in STS at the moment. I'll suggest them to read your paper.
Response: We are not currently studying T-regs subpopulations and would like to see the results of the Milan lab.
Reviewer 2 Report
The manuscript by Gordon and al focuses on one of the most debated topics in sarcoma community: the poor results got with immunotherapy in metstatic STS.
Many attempts have failed to reach a positive results. As consequence immunothrapy in this tumors is not admitted in Europe at least.
Moreover, some attempts to enhace the immunogenity of STS were experimented without concrete results.
The double combination of IPI LIMUMAB + NIVOLUMAB very active in melanoma and moderately active in kidney cancer , associated with Trabectedine, a marine drug drug with a mixed activity of chemotherapic and macrophages inhibitor seems offer somehow posuitive result in this Phase II study.
This paper has a very wide introduction, Materials and method are well described.
The results are positive on PFS and OS , in a classical Phase II trials.
The most impressive results , as before in Twaibi and D'Angelo papers , are reported in LMS, UPS and Dedifferentiated sarcomas.
As before unfortunately no reliable biomarker was found to predict the response to therapy.
In conclusion this study generates interesting hypothesis but a Phas III study comparing IPI + NIVO + TRABE vs DOXO + IFOSFAMIDE is abslutely needed
The manuscript by Gordon and al focuses on one of the most debated topics in sarcoma community: the poor results got with immunotherapy in metstatic STS.
Many attempts have failed to reach some positive results. As consequence immunothrapy in this tumors is not admitted, in Europe at least.
Moreover, some attempts to enhace the immunogenity of STS were experimented without concrete results associating different CPI with different TKI in empirical manner.
The double combination of IPILIMUMAB + NIVOLUMAB very active in melanoma and moderately active in kidney cancer , associated with Trabectedine, a marine drug with a mixed activity of chemotherapic and macrophages inhibitor, seems offer somehow positive result in this Phase II study.
This paper has a very wide introduction, Materials and method are well described.
The results are positive on PFS and OS ,but in a classical Phase II trial.
The most impressive results , as before in Twaibi and D'Angelo papers , are reported in LMS, UPS and Dedifferentiated liposarcomas.
As before, unfortunately no reliable biomarker was found to predict the response to therapy.
In conclusion this study generates interesting hypothesis but a Phase III study comparing IPI + NIVO + TRABE vs DOXO + IFOSFAMIDE is absolutely needed
Author Response
Response to Reviewer 2’s Comments
- The manuscript by Gordon and et al focuses on one of the most debated topics in sarcoma community: the poor results got with immunotherapy in metastatic STS.
Many attempts have failed to reach some positive results. As consequence immunotherapy in these tumors is not admitted, inEurope at least.
Moreover, some attempts to enhance the immunogenity of STS were experimented without concrete results associating different CPI with different TKI in empirical manner.
The double combination of IPILIMUMAB + NIVOLUMAB very active in melanoma and moderately active in kidney cancer, associated with Trabectedine, a marine drug with a mixed activity of chemotherapy and macrophages inhibitor, seems to offer somehow positive result in this Phase II study.
This paper has a very wide introduction, Materials and method are well described.
The results are positive on PFS and OS, but in a classical Phase II trial.
The most impressive results, as before in Twaibi and D'Angelopapers , are reported in LMS, UPS and Dedifferentiated liposarcomas.
As before, unfortunately no reliable biomarker was found to predict the response to therapy.
In conclusion this study generates interesting hypothesis but a Phase III study comparing IPI + NIVO + TRABE vs DOXO +IFOSFAMIDE is absolutely needed.
Response: Thank you for your comments on the SAINT paper. I agree that a Phase III study comparing IPI+NIVO+TRABE vs DOXO+IFOSFAMIde is absolutely needed (See Line 334-335).